# OssCSE: Overcoming Surface Structure Bias in Contrastive Learning for Unsupervised Sentence Embedding

**Zhan Shi**[*]
Bytedance
zhan.shi1@bytedance.com

**Guoyin Wang**
Bytedance
guoyin.wang@bytedance.com

**Ke Bai**
Duke University

**Jiwei Li**
Zhejiang University

**Xiang Li**
Amazon

**Qingjun Cui**
Amazon

**Belinda Zeng**
Amazon

**Trishul Chilimbi**
Amazon

**Xiaodan Zhu**
Queen's University

## Abstract

Contrastive learning has been demonstrated effective in unsupervised sentence representation learning. Given one sentence, positive pairs are obtained by passing the sentence to the encoder twice using the different dropout masks, and negative pairs are obtained by taking another sentence in the same mini-batch. However, the method suffers from the **surface structure bias**, *i.e.*, sentences with similar surface structures will be regarded as close in semantics while sentences with dissimilar surface structures will be viewed as distinct in semantics. This leads to the result that paraphrasing a sentence that is dissimilar in surface structure will receive a lower semantic similarity score than inserting a negative word into the sentence. In this paper, we first verify the bias by collecting a sentence transformation testset. Then we systematically probe the existing models by proposing novel splits based on benchmark datasets in accordance with semantic and surface structure similarity. We tackle the bias in two aspects: balancing the learning target by augmenting with data that counters the bias, and meanwhile preserving word semantics by leveraging recall loss to prevent catastrophic forgetting. We evaluate our model on standard semantic textual similarity (STS) tasks using different pre-trained backbones and achieve state-of-the-art averaged performance across the STS benchmarks. Particularly, our models that are fine-tuned with RoBERTa$_{base}$ and RoBERTa$_{large}$ achieve significantly better performance on most benchmark datasets.

## 1 Introduction

Deep and surface structures (Chomsky, 2009, 2014; Aarts et al., 2018; Xiao et al., 2023) for a linguistic expression, *e.g.*, a sentence, are often used as simple binary opposite terms, with the deep structure representing the semantics, and the surface structure being the actual sentence we see. The deep structure of a linguistic expression is a construct seeking to unify several related surface structures. A particular deep structure can be represented by multiple surface structures, *e.g.*, "*I purchased some beautiful clothes*" and "*Some beautiful clothes were bought by me*" are two surface structures that can be unified by the same deep structure.

Recent studies (Gao et al., 2021; Zhang et al., 2021a,b) show that contrastive learning schemes (Chopra et al., 2005) that are fine-tuned on deep Transformer-based language models pre-trained on large generic corpora, *e.g.*, BERT (Devlin et al., 2018) and RoBERTa (Liu et al., 2019), have significantly benefited the sentence representation learning. The idea of contrastive learning is that positive and negative pairs are constructed given a mini-batch of sentences in training. Specifically, for unsupervised sentence representation learning, the positive pair is composed of two identical sentences and the negative pair is constructed by one sentence with another sentence in the mini-batch. However, this method for forming positive and negative pairs may lead to an inherent bias that the sentences with almost, if not completely, identical surface structures[1] have the same semantic meaning while the sentences with dissimilar surface structures have distinct semantic meanings, leading to the result that semantic meanings are consistently biased towards with surface structure similarities. The bias presents two intermediate questions: (1) To what extent do current models suffer the effects of bias? (2) How to boost the model performance by overcoming the bias?

To answer the first question, we first propose to validate whether current models can correctly rank

---

[*]Work done while interning at Amazon

[1]The inputs are not identical with different dropout masks.

| | DS Sim | Low | High |
|---|---|---|---|
| SS Sim | | | |
| Low | | Cont. | Oppn. |
| High | | Oppn. | Cont. |

Table 1: Two settings (Cont. and Oppn.) by Surface Structure Similarity (SS Sim) and Deep Structure Similarity (DS Sim) for a pair of sentences. Cont. and Oppn. stand for consistence and opposition, respectively.

a few sentence transformations. Furthermore, to systematically evaluate how the bias effect existing models, we split the existing datasets following the Consistency (**Cont.**) and Opposition (**Oppn.**) settings by surface structure and deep structure similarity in Table 1, and then perform evaluations on the two settings accordingly. We find that models with unsupervised training will assign higher scores to negations compared to paraphrases and there is a noticeable difference in performance between Cont. and Oppn. splits.

To answer the second question, since the major bottleneck of current models is their poor performance on the Oppn. datasets, we apply two strategies: (1) automatic sentence-level data augmentations in accordance with the Oppn. setting with max margin loss applied on the augmented data; (2) a regularization loss to prevent catastrophic forgetting in token-level semantics which is critical to constitute sentence meanings. In this work, we use match error rate (MER) (Morris et al., 2004) to measure the surface structure similarity, which is based on the edit distance (Levenshtein et al., 1966). Edit distance is often used for quantitatively capturing the linguistic characteristics of paraphrase pairs (Liu et al., 2022; Androutsopoulos and Malakasiotis, 2010; Magnolini, 2014).

In short, we make the following contributions:

- We investigate the surface structure bias in contrastive learning for the unsupervised sentence representation and systematically evaluate the bias by constructing datasets following the two designated settings on surface structure and deep structure similarity.

- We overcome the bias by leveraging data augmentation according to the Oppn. setting, and then use the max margin loss to incorporate these data in the contrastive learning framework. We also use an additional regularization loss to reduce the catastrophic forgetting in

learning to preserve the word semantics from the pre-trained models.

- Our methods significantly outperform the baselines, achieving the state-of-the-art averaged performance across the benchmark datasets under the standard metric. We provide detailed analyses on how the bias is mediated.

## 2 Related Work

Unsupervised sentence representation learning has been widely studied. The early study attempts to leverage sentence internal structure (Socher et al., 2011; Hill et al., 2016; Le and Mikolov, 2014), and sentence context (Kiros et al., 2015) to learn the representation. Pagliardini et al. (2017) proposes Sent2Vec, a simple unsupervised method that obtains sentence embeddings by combining word vectors with n-gram embeddings. Several efforts have been made to obtain sentence representations using the embedding of some special token or averagely pooled last layer representations, due to the great success of pre-trained language models (Devlin et al., 2018; Liu et al., 2019). However, Ethayarajh (2019) identified the anisotropy problem that the native embeddings from PLMs are concentrated in a small cone in the vector space. To address this issue, techniques such as BERT-flow (Li et al., 2020) and BERT-whitening (Su et al., 2021) have been proposed as post-processing methods.

Recently, unsupervised sentence embeddings have utilized contrastive learning schemes to further boost the performance by different data augmentation methods, such as dropout (Gao et al., 2021; Yan et al., 2021), noise-based negatives (Zhou et al., 2022; Zhang et al., 2022a), deletion-reordering-substitution (Wu et al., 2020), adversarial attack (Yan et al., 2021), span sampling (Giorgi et al., 2021), entity augmentations (Nishikawa et al., 2022) and BERT layer sampling (Kim et al., 2021). In addition to the above research that create pairs of positive and negative representations in different ways, there have been research focusing on improved learning objectives, Zhang et al. (2021b) proposed bootstrapped loss with a moving average of the online network. Zhang et al. (2022b) proposed contrastive loss in angular space. Chuang et al. (2022) added an additional cross entropy loss based on the difference between the original and the transformed sentence. Jiang et al. (2022) leveraged the technology

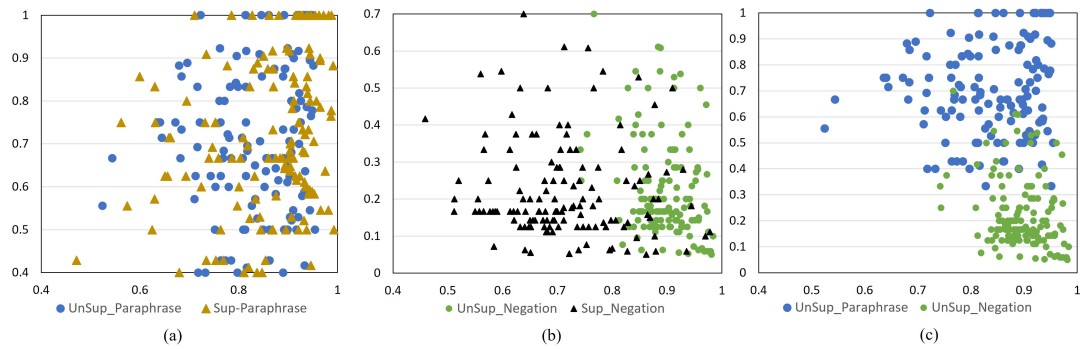

(a)  (b)  (c)

Figure 1: Semantic Similarity Score (Horizontal Axis) and Match Error Rate (Vertical Axis) of all constructed examples. Sub-figure (a) compares the paraphrases by UnSup and Sup models; Sub-figure (b) compares the negations by UnSup and Sup models; Sub-figure (c) compares paraphrases and negations by the UnSup model.

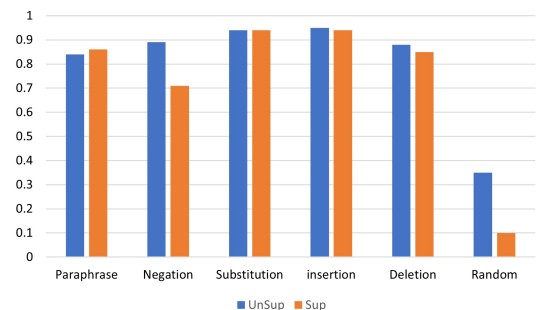

Figure 2: Averaged Semantic Similarity Scores of sentence transformations based on all examples.

of template denoising in the contrastive loss. Although achieving promising results, these methods didn't consider reducing the surface structure bias for contrastive learning in their data augmentation methods or modified learning objectives.

## 3 Probing

### 3.1 Sentence Transformations

To specifically exhibit the bias, we write down seven differently transformed sentences as shown in Example 1 considering six types of sentence transformations: (1) Paraphrasing a sentence with a dissimilar surface forms; (2) Contradicting a sentence by making minor changes; (3) Inserting a few tokens while keeping the meaning unchanged; (4) Deleting a few tokens while keeping the meaning unchanged; (5) Substituting a few tokens while keeping the meaning unchanged; (6) Changing to another random sentence.

We probe the officially released unsupervised and supervised models with Bert-base-uncased as the backbone from SimCSE (Gao et al., 2021)[2]. Each sentence has three associated values: simi-

[2]https://github.com/princeton-nlp/SimCSE

larity values between the original and transformed sentence calculated by both unsupervised and supervised models, and an MER value that measures surface structure similarity.

The MER value (range from $0 \sim 1$) indicates the proportion of words that were incorrectly deleted, substituted and inserted: the lower the value, the higher the surface structure similarity.

$$MER = \frac{S + D + I}{S + D + I + C} \quad (1)$$

where $S$, $D$, $I$ and $C$ are the number of substitutions, deletions, insertions and correct words, respectively.

**Example 1** *Sentence Transformations of "Bryan Cranston will return as Walter White for breaking bad spin off, report claims."*
*(i) Paraphrase: It has been reported that Bryan Cranston will reprise his role as Walter White in a spin-off of Breaking Bad. (UnSup: 0.72, Sup: 0.92, MER: 0.73)*
*(ii) Negation: Bryan Cranston will* not *return as Walter White for Breaking Bad spin off, report claims. (UnSup: 0.96, Sup: 0.75, MER: 0.06)*
*(iii) Deletion: Bryan will return as Walter White for Breaking Bad spin off, report claims. (UnSup: 0.97, Sup: 0.95, MER: 0.07)*
*(iv) Insertion: Bryan Cranston will return as Walter White for breaking bad spin off,* a latest *report claims. (UnSup: 0.95, Sup: 0.98, MER: 0.12)*
*(v) Substitution: Bryan Cranston will* come back *as Walter White for Breaking Bad spin off, report claims. (UnSup: 0.97, Sup: 0.98, MER: 0.13)*
*(vi) Random: Digital era threatens future of drive-ins.(UnSup: 0.40, Sup: 0.08, MER: 1.0)*

**Negation vs. Paraphrase** First of all, Example 1 shows that unsupervised-learning-based (UnSup)

| Datasets | Cont. | Oppn. | Median human scores | Median MER scores |
|---|---|---|---|---|
| STS2012-MSRvid | 406 | 344 | 2.4 | 0.444 |
| STS2013-headlines | 495 | 255 | 2.6 | 0.625 |
| STS2013-OnWN | 286 | 275 | 2.4 | 0.556 |
| STS2014-deft-forum | 251 | 199 | 2.6 | 0.4375 |
| STS2014- headlines | 483 | 267 | 3 | 0.6 |
| STS2014-images | 461 | 289 | 3.2 | 0.5 |
| STS2015-answers-students | 494 | 256 | 3 | 0.75 |
| STS2015-headlines | 517 | 233 | 2.6 | 0.625 |
| STS2015-images | 548 | 202 | 2.5 | 0.538 |
| STS2016-answer-answer | 134 | 120 | 2 | 0.5 |
| STS2016-headlines | 160 | 89 | 2 | 0.625 |
| STS2016-plagiarism | 153 | 77 | 3 | 0.737 |
| STS2016-postediting | 180 | 64 | 3 | 0.416 |
| STS2016-question-question | 75 | 134 | 2 | 0.4 |
| STSB-train | 3532 | 2217 | 3 | 0.536 |
| STSB-test | 855 | 524 | 2.8 | 0.516 |

Table 2: Statistics of the constructed Cont. and Oppn. datasets and splitting criteria.

| | Cont. (UnSup) | Cont. (Sup) | Oppn. (UnSup) | Oppn. (Sup) |
|---|---|---|---|---|
| Weighted Average | 0.8574 | 0.8891 | 0.4982 | 0.6152 |

Table 3: Weighted Average Performance on Cont. and Oppn. datasets by UnSup and Sup models.

models will mistakenly assign higher scores to Negation (0.96) compared to Paraphrase (0.72), indicating that unsupervised-learning-based (Sup) models are biased towards surface structure similarity (MER of the negation is 0.06, which is much lower than the paraphrase: 0.73), while supervised-learning-based models can correctly rank the scores with Negation (0.75) and Paraphrase (0.92). One reason for this is that the Sup model is trained on a natural language inference corpus that uses semantic contradicted sentences to build negative pairs.

**Other Transformations** Secondly, Example 1 also shows that both UnSup and Sup models are able to effectively handle other types of transformations (assigning high similarity scores to sentences with low MER, and low similarity scores to sentences with high MER.), such as Insertion, Deletion, Substitution, and Random. According to Table 1, the above-mentioned four transformations fall under Cont. setting, where surface structure similarity aligns with deep structure similarity. However, Negation and Paraphrase fall under Oppn. setting, where surface structure similarity contradicts deep structure similarity.

### 3.2 Sentence Transformation Collection

To validate our findings on a larger scale, we gather a total of 135 examples and each example shares the same transformations as Example 1. These sentences are from multiple sources, such as Wikipedia, news headlines, and image descriptions. We first use automatic methods and then manually filter these sentences to obtain the final six transformations for each sentence. For paraphrases, we use ChatGPT's (Ouyang et al., 2022) open API by using the template of "*paraphrases of <sentence>*" as the input, and among the multiple outputs, we select the sentence that is the most dissimilar from the original sentence in surface structure. For negations, we collect them by inserting negative words, *e.g.*, *not*, *neither* and *barely*, or replacing words with their antonyms to the original sentences. For the insertions and substitutions, we use Bert-base-uncased as the mask language model to replace or insert around 5% to 10% of the total tokens. For deletion, we randomly remove one word in the sentence. We simply take another sentence in these sentences for the random transformation.

Figure 1 further demonstrates that (1) The UnSup model mistakenly assigns higher scores to negations than paraphrases. (2) The Sup model can correctly assign higher scores to paraphrases than the UnSup model. (3) The Sup model can correctly assign lower scores to the negations than UnSup model. Meanwhile, Figure 2 shows that (1) UnSup and Sup models would give similar average scores in terms of the substitutions, insertions and deletions. (2) Random sentences would be given significantly lower semantic scores. As a result, the major bottleneck between Sup and UnSup models

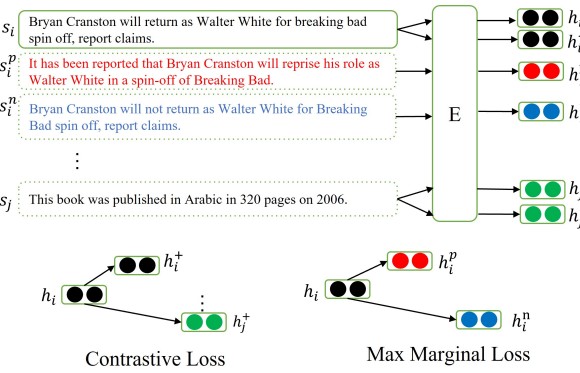

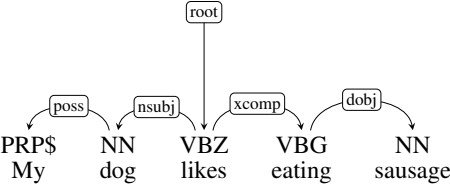

Figure 4: A dependency parsing tree example

lower than that of Cont. (0.8574). On the other hand, the major improvement from the Sup model in comparison to UnSup model is related to Oppn. dataset splits, which achieves an increase of around 12% from 0.4982 to 0.6152 (Refer to Appendix C for detailed results on all splits).

Figure 3: An overview of the augmentation method. $s_i, s_i^p, s_i^n, s_j$ are the original sentence, its paraphrase, its negation, and another sentence in the mini-batch. These sentences are used for different losses.

is mainly related to transformations that fit into the Oppn. setting.

## 4   Method

**Unsupervised Contrastive Learning** Given a dataset of $n$ paired sentences $\{s_i, s_i^+\}_{i=1}^n$, where $s_i$ and $s_i^+$ are semantically similar and regarded as a positive pair. The main idea behind unsupervised contrastive learning is to utilize identical sentences to construct positive pairs. *i.e.*, $s_i^+ = s_i$, and random two sentences in a mini-batch as negative pairs. We feed $s_i$ to the encoder twice by applying different dropout masks in each forward pass and obtaining two sentence embeddings $h_i, h_i^+$ as shown in Figure 3. With $h_i$ and $h_i^+$ for each sentence in a mini-batch with batch size $N$, the contrastive learning objective is formulated as below:

$$l_c = -\log \frac{e^{sim(h_i, h_i^+)/\tau}}{\sum_{j=1}^N e^{sim(h_i, h_j^+)/\tau}} \quad (2)$$

where $\tau$ is a temperature hyper-parameter and $sim$ is the cosine similarity metric.

### 3.3   Dataset Splits

As another step to evaluate the bias, we design new dataset splits based on two settings as illustrated by Table 1. The construction of Cont. and Oppn. dataset splits will help to probe how much the existing models are impacted by the bias.

More specifically, since the human annotation scores are only consistent on each dataset, we split the existing datasets following Cont. and Oppn. settings by surface structure similarity (MER: $0 \sim 1$) and semantic similarity (human annotations: $0 \sim 5$) on the basis of a single dataset as shown in Table 2. Note that all datasets in Table 2 are from the standard benchmark datasets (STS 2012~2016 (Agirre et al., 2012, 2013, 2014, 2015, 2016) and STS-B (Cer et al., 2017)). The table shows the number of Cont. and Oppn. data samples as well as the split criteria (median human annotation and MER scores[3]), *i.e.*, the sentence pairs that are above the median human annotation score and below the median MER score, or below the median human annotation score and above the median MER score would be categorized as the Cont. setting, otherwise Oppn. setting.

We perform probing on each above dataset by the standard spearman's correlation relationship between the model computed scores and human annotated scores. As shown in Table 3, there is a gap in performance between Cont. and Oppn. splits for the UnSup model, and the weighted average performance of Oppn. splits (0.4982) is significantly

Instead of using the original sentence as input and the last hidden layer of *<CLS>* token as sentence representations. Following (Jiang et al., 2022), given any sentence $s$, we use two slightly different prompts to constitute a positive pair.
*Sentence a: This sentence : "$s$" means <mask>*
*Sentence b: This sentence of "$s$" means <mask>*
We use the last hidden layer of the *<mask>* token as the sentence representation. With the prompt, the performance would be higher with lower variance.
**Data augmentation** Since the major bottleneck is the poor performance on the Oppn. dataset splits as the current UnSup models fail to correctly rank the negation and paraphrase transformations, we apply data augmentation based on these two sentence transformations and then leverage a max margin loss between negations and paraphrases so that

---

[3]We only consider that dataset whose median human annotation score is between 2 and 3.5 to remove biased datasets

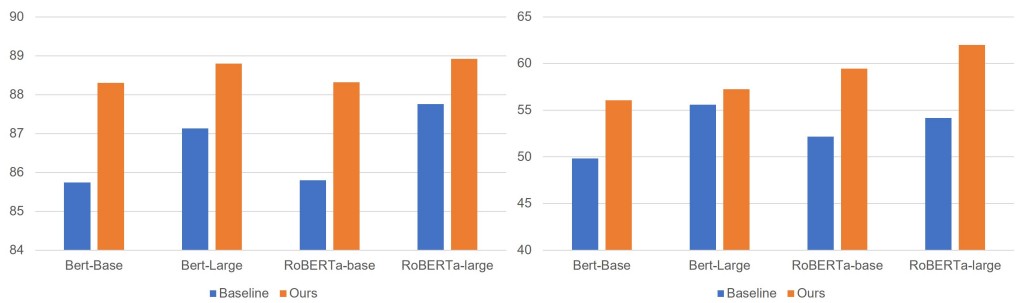

Figure 5: The comparison of averaged performance on Cont. (left) and Oppn. (right) dataset splits.

negations and paraphrases can be ranked properly. Following (Gao et al., 2021), we use the same one million sentences from English Wikipedia for unsupervised learning.

**Step 1**: Generating a semantically contradicted example sentence $s_i^n$ by only inserting negative words to the original sentence $s_i$, *e.g.*, *My dog likes eating sausage $\rightarrow$ My dog does not like eating sausage*. Given the sentence, we use dependency parsing to find the position of the word that is tagged as *root* as shown in Figure 4. Then we insert the negative word based on the pos tag of the *root* word, *e.g.*, if its pos tag is *VBZ*, then we change "likes" $\rightarrow$ "does not like".

**Step 2**: Given a sentence $s_i$, we aim to generate a sentence $s_i^p$ that is similar in semantics but distinctive in surface structure. We use back translation models on English-Russian-English, English-German-English[4] to obtain such candidate sentences. To filter the desired back-translation sentences that satisfy the requirement of surface structure similarity, we then select the examples of which their MER meets the requirement (refer to Appendix A for details)

Hence, we propose the max margin loss based on the $h_i^p$ and $h_i^n$ representations from $s_i^p$ and $s_i^n$:

$$l_m = \max(sim(h_i, h_i^n) - sim(h_i, h_i^p) + \alpha, 0) \\ + \max(sim(h_i, h_i^p) - sim(h_i, h_i^n) - \beta, 0) \quad (3)$$

Although $h_i^n$ is semantically contradictory to the original sentence, it is close to the original one in terms of semantic relatedness. Hence, we design the above two-way max marginal loss, which can guide the model to learn the following objective: $\alpha < sim(h_i, h_i^p) - sim(h_i, h_i^n) < \beta$ where $\alpha, \beta$ are margins.

**Catastrophic forgetting** As shown in Figure 6, we also observe that the training will suffer from

---

[4]https://huggingface.co/facebook/wmt19-ru-en,wmt19-en-ru,wmt19-de-en,wmt19-en-de

performance degradation on Oppo. dataset splits after 3,000 training steps (0.3 epoch). It might be due to the catastrophic forgetting of word semantics and Oppn. splits are more relied on word semantics rather than surface structure. However, the fine-tuning process with pre-trained language models is prone to catastrophic forgetting, and it will cause a significant drop in performance on the Oppn. dataset splits particularly. We use the learning objective from (Chen et al., 2020), which proposes a pretraining simulation mechanism to recall the knowledge from pretraining tasks $D_S$ without data.

$$l_r = -\log(\theta|D_S) \approx \frac{1}{2}\gamma \sum_i (\theta_i - \theta_i^*)^2 \quad (4)$$

where $\gamma$ is a hyper-parameter for this recall loss and $\theta^*$ is the initial parameters of the pre-trained models. (Refer to Appendix D for background of the recall loss). Overall, the loss function for our method is as below:

$$\mathcal{L} = l_c + l_r + \lambda l_m \quad (5)$$

## 5 Experiments

### 5.1 Experiment Setup

We perform experiments with backbones of RoBERTa$_{base}$, RoBERTa$_{large}$ (Liu et al., 2019), BERT$_{base}$, and BERT$_{large}$ (Devlin et al., 2018). We use semantic textual similarity tasks: STS 2012–2016 (Agirre et al., 2012, 2013, 2014, 2015, 2016), STS Benchmark (Cer et al., 2017) and SICKRelatedness (Marelli et al., 2014). STS12-STS16 datasets do not have train or development sets, and thus we evaluate the models on the development set of STS-B to search for better settings of the hyper-parameters. We also evaluate our method on the 7 transfer learning benchmark tasks (MR (Pang and Lee, 2005), CR (Hu and Liu, 2004), SUBJ (Pang and Lee, 2004), MPQA (Wiebe et al., 2005), SST-2 (Socher et al.,

| Method | STS12 | STS13 | STS14 | STS15 | STS16 | STS-B | SICK-R | Avg. |
|---|---|---|---|---|---|---|---|---|
| SimCSE-RoBERTa$_{base}$ | 70.16 | 81.77 | 73.24 | 81.36 | 80.65 | 80.22 | 68.56 | 76.57 |
| DiffCSE-RoBERTa$_{base}$ | 70.05 | 83.43 | 75.49 | 82.81 | 82.12 | 82.38 | 71.19 | 78.21 |
| DCLR-RoBERTa$_{base}$ | 70.01 | 83.08 | 75.09 | 83.66 | 81.06 | 81.86 | 70.33 | 77.87 |
| ESimCSE-RoBERTa$_{base}$ | 69.90 | 82.50 | 74.68 | 83.19 | 80.30 | 80.99 | 70.54 | 77.44 |
| Prompt-RoBERTa$_{base}$ | **73.94** | 84.74 | 77.28 | **84.99** | 81.74 | 81.88 | 69.50 | 79.15 |
| PCL-RoBERTa$_{base}$ | 71.13 | 82.38 | 75.40 | 83.07 | 81.98 | 81.63 | 69.72 | 77.90 |
| (Ours)-OssCSE RoBERTa$_{base}$ | 72.28 | **85.27** | **79.51** | 84.77 | **82.32** | **83.55** | **75.54** | **80.46** |
| SimCSE-BERT$_{base}$ | 68.40 | 82.41 | 74.38 | 80.91 | 78.56 | 76.85 | 72.23 | 76.25 |
| ArcCSE-BERT$_{base}$ | 72.08 | 84.27 | 76.25 | 82.32 | 79.54 | 79.92 | 72.39 | 78.11 |
| DiffCSE-BERT$_{base}$ | 72.28 | 84.43 | 76.47 | 83.90 | 80.54 | 80.59 | 71.23 | 78.49 |
| DCLR-BERT$_{base}$ | 70.81 | 83.73 | 75.11 | 82.56 | 78.44 | 78.31 | 71.59 | 77.22 |
| ESimCSE-BERT$_{base}$ | **73.40** | 83.27 | 77.25 | 82.66 | 78.81 | 80.17 | 72.30 | 78.27 |
| Prompt-BERT$_{base}$ | 71.56 | 84.58 | 76.98 | 84.47 | 80.60 | 81.60 | 69.87 | 78.54 |
| InfoCSE-BERT$_{base}$ | 70.53 | **84.59** | 76.40 | **85.10** | **81.95** | **82.00** | 71.37 | 78.85 |
| PCL-BERT$_{base}$ | 72.84 | 83.81 | 76.52 | 83.06 | 79.32 | 80.01 | 73.38 | 78.42 |
| (Ours)-OssCSE BERT$_{base}$ | 71.78 | 84.40 | **77.71** | 83.95 | 79.92 | 80.57 | **75.25** | **79.08** |
| SimCSE-RoBERTa$_{large}$ | 72.86 | 83.99 | 75.62 | 84.77 | 81.80 | 81.98 | 71.26 | 78.90 |
| DCLR-RoBERTa$_{large}$ | 73.09 | 84.57 | 76.13 | 85.15 | 81.99 | 82.35 | 71.8 | 79.30 |
| ESimCSE-RoBERTa$_{large}$ | 73.2 | 84.93 | 76.88 | 84.86 | 81.21 | 82.79 | 72.27 | 79.40 |
| (Ours)-OssCSE RoBERTa$_{large}$ | **74.56** | **85.54** | **79.88** | **86.64** | **82.10** | **84.42** | **78.65** | **81.68** |
| SimCSE-BERT$_{large}$ | 70.88 | 84.16 | 76.43 | 84.50 | 79.76 | 79.26 | 73.88 | 78.41 |
| ArcCSE-BERT$_{large}$ | **73.17** | 86.19 | 77.90 | 84.97 | 79.43 | 80.45 | 73.50 | 79.37 |
| DCLR-BERT$_{large}$ | 71.87 | 84.83 | 77.37 | 84.70 | 79.81 | 79.55 | 74.19 | 78.90 |
| ESimCSE-BERT$_{large}$ | 73.21 | 85.37 | 77.73 | 84.30 | 78.92 | 80.73 | 74.89 | 79.31 |
| InfoCSE-BERT$_{large}$ | 71.89 | 86.17 | 77.72 | **86.20** | **81.29** | **83.16** | 74.84 | 80.18 |
| (Ours)-OssCSE BERT$_{large}$ | 72.64 | **86.36** | **79.16** | 85.04 | 80.80 | 82.61 | **76.65** | **80.47** |

Table 4: The performance comparison of our methods with different backbones on sentence similarity tasks. The performance is based on the default random seed. The baseline results are from their published papers.

| Method | MR | CR | SUBJ | MPQA | SST | TREC | MRPC | Avg. |
|---|---|---|---|---|---|---|---|---|
| SimCSE-RoBERTa$_{base}$ | 81.04 | 87.74 | 93.28 | 86.94 | 86.60 | 84.60 | 73.68 | 84.84 |
| SimCSE-RoBERTa$_{base}$ + Prompt | 81.01 | 86.85 | 93.45 | 88.42 | 84.90 | 87.10 | 74.36 | 85.16 |
| (Ours)-OssCSE RoBERTa$_{base}$ | **82.88** | **88.13** | **93.53** | **90.54** | **87.26** | **88.80** | **77.97** | **87.02** |

Table 5: The performance comparison of our methods with RoBERTa-base on the transfer tasks.

2013), TREC (Voorhees and Tice, 2000) and MRPC (Dolan and Brockett, 2005)). The SentEval toolkit is used for evaluation, and the Spearman correlation coefficient is used as the standard evaluation metric. We also perform experiments on the dataset splits constructed by the Cont. and Oppn. settings in Table 2 to validate whether surface structure bias has been alleviated. Refer to Appendix A for implementation details.

## 5.2 Results and Analyses

**Compared Baselines** We mainly choose SimCSE for comparison, since we build our method based on it and shares the same setting in our approach. We also compare our results with several most recent strong work as below: PromptBert (Jiang et al., 2022), ArcCSE (Zhang et al., 2022b), DiffCSE (Chuang et al., 2022), InfoCSE (Wu et al., 2022b), ESimCSE (Wu et al., 2021), DCLR (Zhou et al., 2022), and PCL (Wu et al., 2022a).

**Sentence Similarity Tasks** We show the results of STS tasks in Table 4. OssCSE fine-tuned with different pre-trained models can significantly outperform all the baselines, particularly with RoBERTa$_{base}$ and RoBERTa$_{large}$. OssCSE-RoBERTa$_{base}$ raises the averaged Spearman's correlation from 76.57% to 80.46% compared with SimCSE-RoBERTa$_{base}$, and OssCSE-RoBERTa$_{large}$ increases the averaged Spearman's correlation from 78.90% to 81.68% compared with SimCSE-RoBERTa$_{base}$. Meanwhile, for BERT$_{base}$ and BERT$_{large}$, our method can also improve upon SimCSE-BERT$_{base}$ and SimCSE-BERT$_{large}$ significantly, with an improvement of 2.8% and 2.5% respectively. The reason why our approach achieves greater improvement on RoBERTa than BERT backbones is that RoBERTa has been demonstrated to have learned better word representations(Liu et al., 2019), and thus making the recall loss more effective on the RoBERTa

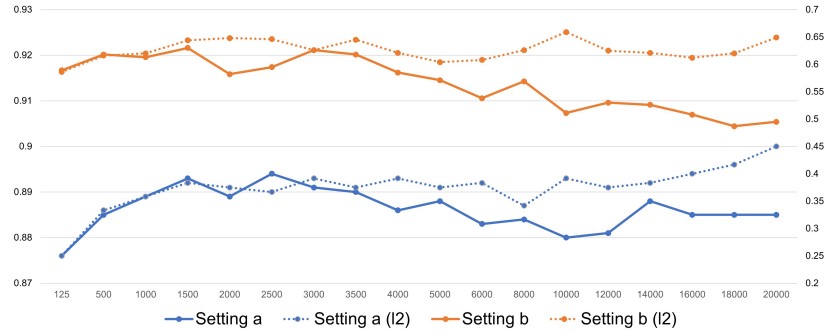

Figure 6: Training checkpoints on Cont. and Oppn. splits of STSB-dev, w and w/o the recall loss (l2).

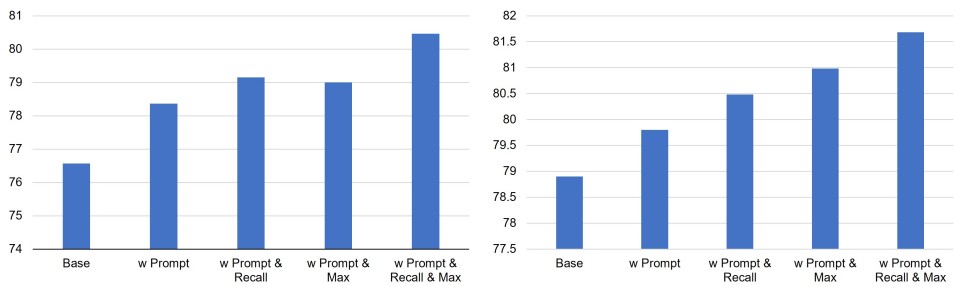

Figure 7: Ablation analysis of averaged performance on the seven sentence similarity datasets with RoBERTa-base (left) and RoBERTa-large (right) as backbones, where max denotes max margin loss.

backbones.

**Transfer Tasks** We show the results of transfer tasks in Table 5 with RoBERTa$_{base}$ as backbone. Compared with SimCSE-RoBERTa$_{base}$, our approach can improve the averaged score from 84.84% to 87.02%. We find that using recall loss alone can greatly enhance performance on tasks involving transfer learning. (85.06% to 87.12%), confirming that catastrophic forgetting prevention objectives can also benefit transfer tasks.

**Cont. and Oppn.** We show the results of Cont. and Oppn. dataset splits in Figure 5. The figure indicates that there are performance boost with all the four backbones. We find that the improvement on Oppn. is more significant, specifically when using the backbones of RoBERTa$_{base}$ and RoBERTa$_{large}$, achieving an averaged improvement of around 8%. The improvement on Cont. dataset splits is relatively smaller, with an averaged increase of around 2.5% on the all backbones.

**Ablation Analysis** Figure 7 shows the ablation of averaged performance with the backbones of RoBERTa$_{base}$ and RoBERTa$_{large}$. In OssCSE, both the recall loss and the max margin loss are crucial because they consistently improve our method. Take RoBERTa$_{base}$ for instance, the averaged performance decreases from 80.46% to around 79% if the recall loss is removed. Similarly, if the max

margin loss is removed, the average performance drops from 80.46% to around 79.2%. This result shows the importance to have all designated loss items that exist together in the learning objective. We also find that recall loss can help stabilize the learning process because of its effectiveness in preventing catastrophic forgetting. Figure 6 shows find that the performance on the Oppn. split of STSB-dev becomes stable when recall loss is used.

**Quantitative Analysis** We further take the original sentence in Example 1, and write down eight paraphrases and negations (Refer to Appendix B for the specific sentence transformations). When comparing OssCSE-BERT$_{base}$ with SimCSE-BERT$_{base}$, we find that the averaged score of eight paraphrases increases from 0.79 to 0.88 and the averaged score of eight negations decreases from 0.93 to 0.87.

## 6 Conclusion

We investigate the surface structure bias in contrastive learning for the unsupervised sentence embedding and systematically probe the bias by constructing datasets following the two designated settings on the surface and deep structure similarity. We overcome the bias by data augmentation methods and then use the max margin loss to incorporate these data in the contrastive learning framework. We also use a recall loss to reduce catastrophic

forgetting in unsupervised learning to preserve the word semantics in the pre-trained models. The results significantly outperform the baselines and achieve state-of-the-art results on averaged performance with different pre-trained backbones.

# 7 Limitations

First, we cannot guarantee the quality of the back-translation results as the augmentation target of paraphrase. There might be error propagation in the forward and backward translation process. We are considering using strong paraphrase models in the future. Second, we have not considered other minor modification methods that would be considered significant to semantic meanings rather than negations. Missing these parts may weaken our model generalization ability such that it may not be applicable in specific domains.

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

## A  Implementation Details

To make fair comparison, we use exactly the same settings with SimCSE except descriptions below. We train the UnSup models for two epoch. We carry out the batch size of 64 for all backbones, and learning rate of 1e-5 for $BERT_{base}$ and $RoBERTa_{base}$, and 5e-6 for $RoBERTa_{large}$ and $BERT_{large}$. We validate evey 125 steps on STS-B development set and adopt the following hyper-parameter settings and data augmentation criteria: (1) We use the following hyper-parameters for the two-way max margin loss: $\alpha = 0.05, \beta = 0.2, \lambda = 1e - 3$ and recall loss item ($\gamma = 2e - 3$), and we find our method achieves the best performance on the STSB-dev set using the above parameters through grid search; (2) We select the back translation results of which the MER are within 0.15 to 0.6 as the criteria of the paraphrase transformations. We don't use the sentences that have even higher MER values because these sentences may have been corrupted by the translation errors.

## B  Example

**Example 2** *Sentence Transformations of "bryan cranston will return as walter white for breaking bad spin off, report claims."*
*(i) Paraphrase:*
*(1) report says bryan cranston will be back to act walter white as breaking bad sequel.*
*(2) in breaking bad sequel, the actor of walter white, bryan cranston, will return.*
*(3) breaking bad will still use bryan cranston as the actor of walter white in its spin off.*
*(4) walter white will continue to be acted by bryan cranston in its spin off*
*(5) the actor of walter white, i.e., bryan cranston, in breaking bad spin off will keep unchanged*
*(6) bryan cranston is reported to return for the breaking bad sequel as the actor of walter white*
*(7) the role of walter white will continue to be acted by bryan cranston, according to the report*
*(8) the report says the breaking bad spin off will still have bryan cranston played same role, walter white*
*(ii) Negation:*
*(1) bryan cranston will not return as walter white for breaking bad spin off , report claims*
*(2) bryan cranston will return as walter white for breaking bad spin off , fake report claims*
*(3) bryan cranston is unlikely to return as walter white for breaking bad spin off , report claims*
*(4) bryan cranston cannot return as walter white for breaking bad spin off , report claims*
*(5) bryan cranston impossible to return as walter white for breaking bad spin off , report claims*
*(6) bryan cranston never return as walter white for breaking bad spin off , report claims*
*(7) bryan cranston return as walter white for breaking bad spin off , report falsely claims*
*(8) bryan cranston less likely to return as walter white for breaking bad spin off , report claims*

## C  Performance on Cont. and Oppn. splits

All results of the Cont, and Oppn. dataset splits are shown in Table 6.

## D  Recall Loss

$$
\begin{aligned}
l_r &= \log(\theta|D_S) \\
&\approx \frac{1}{2}(\theta - \theta^*)^T H(\theta^*)(\theta - \theta^*) \\
&\approx \frac{1}{2}(\theta - \theta^*)^T (NF(\theta^*) + H_{prior}(\theta^*))(\theta - \theta^*) \\
&\approx \frac{1}{2} N \sum_i F_i (\theta_i - \theta_i^*)^2 \qquad (6) \\
&\approx \frac{1}{2} N F \sum_i (\theta_i - \theta_i^*)^2 \\
&= \frac{1}{2} \gamma \sum_i (\theta_i - \theta_i^*)^2
\end{aligned}
$$

| Datasets | Cont. (UnSup) | Cont. (Sup) | Oppn. (UnSup) | Oppn. (Sup) |
|---|---|---|---|---|
| STS2012-MSRvid | 0.8848 | 0.9388 | 0.7306 | 0.8862 |
| STS2013-headlines | 0.8688 | 0.881 | 0.4487 | 0.5281 |
| STS2013-OnWN | 0.8584 | 0.8788 | 0.7816 | 0.814 |
| STS2014-deft-forum | 0.7269 | 0.7942 | 0.355 | 0.3279 |
| STS2014- headlines | 0.8578 | 0.8898 | 0.4066 | 0.364 |
| STS2014-images | 0.8723 | 0.9108 | 0.3638 | 0.645 |
| STS2015-answers-students | 0.8394 | 0.8264 | 0.322 | 0.3992 |
| STS2015-headlines | 0.8932 | 0.9081 | 0.402 | 0.4602 |
| STS2015-images | 0.8926 | 0.9446 | 0.5648 | 0.8224 |
| STS2016-answer-answer | 0.7646 | 0.841 | 0.5011 | 0.607 |
| STS2016-headlines | 0.8887 | 0.8793 | 0.3836 | 0.4075 |
| STS2016-plagiarism | 0.9098 | 0.9137 | 0.5058 | 0.5425 |
| STS2016-postediting | 0.8982 | 0.9064 | 0.0719 | 0.0835 |
| STS2016-question-question | 0.7187 | 0.7517 | 0.6347 | 0.6821 |
| STSB-train | 0.8577 | 0.8928 | 0.4966 | 0.6293 |
| STSB-test | 0.8354 | 0.8762 | 0.5415 | 0.7271 |
| Weighted average | 0.8574 | 0.8891 | 0.4982 | 0.6152 |

Table 6: All Performance on Cont. and Oppn. dataset splits when probed by unsupervised and supervised models.