# OpenReview forum: "OssCSE: Overcoming Surface Structure Bias in Contrastive Learning for Unsupervised Sentence Embedding"
_EMNLP/2023/Conference — EMNLP 2023 Main_

### Official Review · Reviewer_w2Dv · 2023-07-22

**Typos Grammar Style And Presentation Improvements:** None
**Soundness:** 4

**Excitement:**

4: Strong: This paper deepens the understanding of some phenomenon or lowers the barriers to an existing research direction.

**Missing References:**

Missing the latest SOTA[1], which was first uploaded and submitted to OpenReview last September, which cannot be considered contemporaneous work. The STS results in this paper are almost equivalent to the latest SOTA.

[1] RankCSE: Unsupervised Sentence Representations Learning via Learning to Rank

**Paper Topic And Main Contributions:**

This paper finds and discusses the surface structure bias problem in SimCSE, an unsupervised sentence representation learning baseline method.

The solution in the paper is to obtain the "Paraphrase" and "Negation" forms of some sentences by data augmentation methods, then two triplet losses are used to keep a gap between similarity(original sentences, paraphrase form) and the similarity(original sentences, negation form). In addition, a regression loss is used to solve the catastrophic forgetting problem during training.

**Questions For The Authors:**

Although there is no problem with the general logic of the paper in finding surface structure bias and proposing a method to solve it, I think that one of the points made by the author during the discussion has not been verified:

Question1. The authors wrote "This method for forming positive and negative pairs may lead to the surface structure bias" in the paper. However, BERT-Flow shows that this problem already exists in the original BERT. However, the probing experiments in Section 3 of the paper do not report results from the original BERT model, so I don't think it can be said that the method of SimCSE for constructing positive and negative examples introduces this problem, probably because SimCSE doesn't (fully) address this problem that originally existed in the BERT-based Model.

**Reasons To Accept:**

1、The problem of highly correlated semantic similarity and edit distance that exists in BERT was found in BERT-Flow[1] a long time ago, which is called "surface structure bias" in this paper. However, after the unsupervised method based on contrastive learning has been proposed, this problem has not been revisited and verified whether it has been solved or not, and I think this work is valuable.

2、 The logic of this paper is perfect,  it fully reveals to us the surface structure bias problem existing in SimCSE, and the experiments do prove that the method does effectively alleviate the bias problem.

[1] On the Sentence Embeddings from Pre-trained Language Models. EMNLP2020

**Reasons To Reject:**

The experimental part still has some flaws:

1. The authors used SimCSE as the research object in all of the previous exploratory experiments. However, the "prompt trick" in PromptBERT becomes the fixed part to be added to the Experiment part.  I think the ablation study should include at least three experiments, "w Max", "w Recall" and "w Recall & Max", to illustrate the effectiveness of the purposed method directly on SimCSE.
2.  According to my personal experience, random seeds have some influence on the experimental results. Referring to the previous work [1,2,3], reporting the average results of experiments with 3 or more random seeds is more convincing.
3. The paper does not state how many new sentences were involved in the training process on top of the 1M sentences. It is important to know whether the improvement of this method is simply from the introduction of more data compared to those methods that only perform hard negative sample mining or new loss functions.
4. Insufficient exploration of hyperparameters, neither the search range nor the degree of influence of hyperparameters is described in the paper.

Besides, the authors did not commit to open-source code with the constructed probe dataset, which makes the results potentially difficult to reproduce.

I will be glad to see the author's response!


[1] Unsupervised Sentence Representation via Contrastive Learning with Mixing Negatives. AAAI2022

[2] PCL: Peer-Contrastive Learning with Diverse Augmentations for Unsupervised Sentence Embeddings. EMNLP2022

[3] On The Inadequacy of Optimizing Alignment and Uniformity in Contrastive Learning of Sentence Representations. ICLR2023

**Reproducibility:**

3: Could reproduce the results with some difficulty. The settings of parameters are underspecified or subjectively determined; the training/evaluation data are not widely available.

**Reviewer Confidence:**

5: Positive that my evaluation is correct. I read the paper very carefully and I am very familiar with related work.

---

> ### Author Rebuttal · Authors · 2023-08-28
>
> 1. Ablation analysis on vanilla SimCSE: Thank you for your advice, here are the results on BERT-base:
>
>         STS12 | STS13 | STS14 | STS15 | STS16 | STS-B | SICK-R | Avg.
>
> SimCSE                  68.40 | 82.41 | 74.38 | 80.91 | 78.56 | 76.85 | 72.23 | 76.25
>
> Ours w Max             72.77 | 83.92 | 76.84 | 82.57 | 79.53 | 78.67 | 73.76 | 78.29
>
> Ours w Rec             69.20 | 82.19 | 74.16 | 82.96 | 80.60 | 79.50 | 72.86 | 77.35
>
> Ours w Max & Rec  70.61 | 85.48 | 76.39 | 84.59 | 79.91 | 80.12 | 74.99 | 78.87
>
> The performance gain from leveraging max margin loss and recall loss based on vanilla SimCSE is similar to using PromptBERT as the backbone.
>
>
> 2. Number of augmented samples: Thank you for your question. I agree with you that augmented data is important for the final performance. According to the augmentation method from Appendix A, there are around 590,000 augmented samples (overall number of training examples is 1 million) that have both paraphrase and negation transformations. Inspired by your suggestions, we add three ablation results by using 50,000, 100,000 and 300,000 on Roberta base. Here are the Avg performances of the seven standard benchmarks.
> 50,000 -> 79.78;
> 100,000 -> 80.21;
> 300,000 -> 80.49;
> 590,000 -> 80.46;
> We find that using 100,000 augmented samples can basically achieve similar performance (80.21) to our reported result (80.46), and using 300,000 (80.49) is even better than our reported result (80.46).
>
> 3. Random seeds effect: Thank you for your suggestions, I agree with you on the random seed effect on the final performance, and all our reported results on based on the default seed (42). One reason why we choose PromptBERT as part of the overall method is that its variance in the results is smaller than vanilla SimCSE. We rerun our methods on Roberta-base experiments by setting 5 random seeds, and we find the Avg. performance of Roberta-base ranges from [80.36, 80.55]. We will report the variance in the final results when revision.
>
> 4. Hyperparameter search and code release: Thank you for your constructive comments. The hyperparameters are revealed in Appendix A. We will report the hyperparameters selection experimental results when revision. The code and the trained model links have been provided in the supplemental materials. We will release our trained model and code once the paper is accepted.
>
> 5. Bias in the original BERT: I agree with you that the original BERT might already have such bias according to the BERT-flow paper, and we will re-verify this in the revision. From the perspective of the recent LLMs, slightly supervised fine-tuning after the unsupervised pretraining stage is always necessary to learn the sentence-level meaning, also known as "instruction following ability". We will consider that as a possible way to enhance this work in the future.
>
> 6. Missing References: Thank you for pointing this out, we will include the RankCSE paper in the revision.

---

### Official Review · Reviewer_u5Vf · 2023-08-03

**Soundness:** 3

**Excitement:**

3: Ambivalent: It has merits (e.g., it reports state-of-the-art results, the idea is nice), but there are key weaknesses (e.g., it describes incremental work), and it can significantly benefit from another round of revision. However, I won't object to accepting it if my co-reviewers champion it.

**Paper Topic And Main Contributions:**

This paper addresses the issue of the positive and negative pairs formation for learning sentence representations with unsupervised contrastive learning. The problem is that two sentences different in structures (they call surface structures) may end up sharing the same semantic meanings, which they call similar deep structures. The authors tried to alleviate this issue with a novel objective for learning augmented data and a regularization technique for the pre-trained model.

**Questions For The Authors:**

(A) Why do we need to restrict the learning objective in Eq (3) with margins?

(B) Have you tried to add the synthetic positive and negative pairs into the original contrastive learning objective, instead of creating a new objective of max margin loss?

**Reasons To Accept:**

The authors provide rigorous and extensive experiments to address the problem of surface structure bias, showing its importance to unsupervised sentence embedding approaches. In addition, compared with several baselines, the proposed method seems to be effective, especially the enhancement on RoBERTa series models.

**Reasons To Reject:**

The overall method in this paper uses the proposed method with the prompt technique, which is not related to the issue of structure bias. The prompt technique is supposed to serve as an amplifier. This causes two problems:

(A) Many baselines did not use prompts, but the authors's approach did. The authors compare their approach with the baseline with prompt (SimCSE-RoBERTa_{base}+Prompt) in Table 5 on the NLU tasks, but they did not compare the proposed method with the setting in Table 4 on the similarity tasks, which are the main tasks in this paper.

(B) It's unclear if the prompt technique plays a more important role over the proposed method. An ablation study may be required, but all the settings on top of "Base" in Figure 7 include prompts.

**Reproducibility:**

4: Could mostly reproduce the results, but there may be some variation because of sample variance or minor variations in their interpretation of the protocol or method.

**Reviewer Confidence:**

4: Quite sure. I tried to check the important points carefully. It's unlikely, though conceivable, that I missed something that should affect my ratings.

**Typos Grammar Style And Presentation Improvements:**

(A) The paper is hard to follow. For example, when Cont. and Oppn. first show up in L080, I expect to see clear definitions of Cont. and Oppn. with examples in the following lines or in Table 1, but only short indications in the caption of Table 1. And Figure 6. is mentioned at the left column of Page 6 but not show up until the last page.

(B) L392 typo: Oppo. -> Oppn.

(C) The legend of Figure 6. can be improved to match the caption.

---

> ### Author Rebuttal · Authors · 2023-08-28
>
> 1. Why use PromptBert?
> Thank you for your question. As highlighted by Reviewer w2Dv, random seeds can influence the experimental outcomes in this unsupervised learning task, particularly with the vanilla SimCSE. However, PromptBert offers a more consistent training process and yields superior, more consistent results by leveraging a prompt combined with the “[mask]” token for sentence representation, building upon the foundation of SimCSE. This approach has been adopted in several recent sentence embedding studies [1] [2]. Our experiments, based on PromptBert, show significantly less variance compared to that on SimCSE.
> [1] Yeon et al. Ranking-Enhanced Unsupervised Sentence Representation Learning. In Proceedings of ACL 2023
> [2] Zeng, Jiali, et al. "Contrastive Learning with Prompt-derived Virtual Semantic Prototypes for Unsupervised Sentence Embedding." Findings of the Association for Computational Linguistics: EMNLP 2022.
>
> 2. Ablation analysis on vanilla SimCSE (BERT-base): Thank you for your advice, here are the results:
>
>         STS12 | STS13 | STS14 | STS15 | STS16 | STS-B | SICK-R | Avg.
>
> SimCSE                  68.40 | 82.41 | 74.38 | 80.91 | 78.56 | 76.85 | 72.23 | 76.25
>
> SimCSE w Max             72.77 | 83.92 | 76.84 | 82.57 | 79.53 | 78.67 | 73.76 | 78.29
>
> SimCSE w Rec             69.20 | 82.19 | 74.16 | 82.96 | 80.60 | 79.50 | 72.86 | 77.35
>
> SimCSE w Max & Rec  70.61 | 85.48 | 76.39 | 84.59 | 79.91 | 80.12 | 74.99 | 78.87
>
> The performance gain from leveraging max margin loss and recall loss based on vanilla SimCSE is similar to using PromptBERT as the backbone.
>
> 3. Directly adding the augmented samples in SimCSE:
> Thank you for your question. We tried directly utilizing these sentences as positive and negative samples in order to fit them into contrastive learning objectives. Our experiments indicate a substantial decrease in average performance from 80.46 to 75.18 when utilizing Roberta-base as the backbone. The contrastive loss is particularly sensitive to the quality of generated paraphrases. This is consistent with the findings presented in Table 1 and 4 of the SimCSE paper, where performance decreases when applying augmented or even labeled (such as image caption datasets) data in contrastive loss.
>
> 4. Margins of Max Margin Loss:
> Thank you for your question. Given this sensitivity of data quality, we have set alpha (0.05), which can guarantee the correct ranking between paraphrases and negations. Furthermore, a large alpha could increase the possibility of poorly generated paraphrases dominating the training.
> Secondly, it is not appropriate to view negation sentences as hard negatives. These sentences are semantically contradictory, yet still highly semantically related to the original sentence (usually a single-word change). As such, we have set a value for beta (0.2) to constrain these negations to also share a relatively high level of sentence similarity.
>
> 5. Typos, structures, and grammar errors: Thank you for your comment, I will improve the readability of the paper and correct the typos and grammar errors in revision.

---

### Official Review · Reviewer_ApYm · 2023-08-05

**Soundness:** 4

**Excitement:**

4: Strong: This paper deepens the understanding of some phenomenon or lowers the barriers to an existing research direction.

**Missing References:**

The extremely high intra-sentence similarity phenomenon brought by supervised contrastive learning [1] might be relavant to the phenomenon found in the paper (might not). Using dropout-based unsupervised methods, there's no lexical differences to provide such signals for the models to learn to rely on important words (and thus provide high intra-sentence similarity), such that the models learn to rely on "surface-level structure". This gap might particularly contribute to the "random" part.

[1] On Isotropy, Contextualization and Learning Dynamics of Contrastive-based Sentence Representation Learning. In Findings of the Association for Computational Linguistics: ACL 2023.

**Paper Topic And Main Contributions:**

The paper identifies two classes of text (negations and random) that contribute the most to the performance gap between unsupervised contrastive learning models and their supervised counterparts. The authors attribute the problems to that unsupervised methods learn surface-level structures, and propose methods accordingly to bridge the gap, providing good unsupervised performance.

**Questions For The Authors:**

1. See reasons to reject.

2. In the ablation analysis, it seems that the "base" means original simcse baselines, and the authors ablate prompt, recall loss and the max margin loss components. I personally think the data that was generated certainly contributes a lot to the performance gain as well (particularly the back-translation part, because this is essentially positive pairs generated by other supervised models), and should be emphasized, instead of attributing the gain to max margin loss. Could the authors further justify this?

3. Do unsupervised methods other than dropout-based methods have the same gap with supervised methods on negations and random?

**Reasons To Accept:**

The paper is well-written. The authors are very familiar with related work. The motivations are clear, and certainly not incremental to the field. Identifying the problems first then propose methods accordingly is a good way to approach the gap.

**Reasons To Reject:**

The paper uses many techniques, partly making the framework not too targeted towards addressing its central claim (that surface structure bias contributes a lot to the performance degrade of unsupervised methods). For instance, using PromptBERT instead of [cls] pooling seems to come out of nowhere (also, have you tried mean pooling)? Other than these, I don't see many reasons to reject.

**Reproducibility:**

4: Could mostly reproduce the results, but there may be some variation because of sample variance or minor variations in their interpretation of the protocol or method.

**Reviewer Confidence:**

4: Quite sure. I tried to check the important points carefully. It's unlikely, though conceivable, that I missed something that should affect my ratings.

---

> ### Author Rebuttal · Authors · 2023-08-28
>
> 1. Why use PromptBert?
> Thank you for your question. As highlighted by Reviewer w2Dv, random seeds can influence the experimental outcomes in this unsupervised learning task, particularly with the vanilla SimCSE. However, PromptBert offers a more consistent training process and yields superior, more consistent results by leveraging a prompt combined with the “[mask]” token for sentence representation, building upon the foundation of SimCSE. This approach has been adopted in several recent sentence embedding studies [1] [2]. Our experiments, based on PromptBert, show significantly less variance compared to that on SimCSE.
> [1] Yeon et al. Ranking-Enhanced Unsupervised Sentence Representation Learning. In Proceedings of ACL 2023
> [2] Zeng, Jiali, et al. "Contrastive Learning with Prompt-derived Virtual Semantic Prototypes for Unsupervised Sentence Embedding." Findings of the Association for Computational Linguistics: EMNLP 2022.
>
> 2. How does the augmented data contribute to the performance?
> Thank you for your question. According to the augmentation method from Appendix A, there are around 590,000 augmented samples (out of 1 million total samples) that have both paraphrase and negation transformations. Inspired by your suggestions, we add three ablation results by using 50,000, 100,000 and 300,000 on Roberta base.
> Here are the Avg performances of the seven standard benchmarks.
> 50,000 -> 79.78; 100,000 -> 80.21; 300,000 -> 80.49; 590,000 -> 80.46 (This work).
> We find that using 100,000 augmented samples can basically achieve similar performance (80.21) to our reported result (80.46), and using 300,000 (80.49) is even better than our reported result (80.46). We will include this experiments in revision.
>
> 3. Do other unsupervised methods also suffer from the bias?
> Thank you for your inspiring question. From the perspective of current LLMs, pretrained models, either BERT or GPT-like, require a supervised fine-tuning period to learn the sentence-level meaning, maybe also called “instruction following ability”.  I believe we can significantly reduce the number of augmented samples by using stronger backbones and high quality samples. We will leave it for future work.
>
> 4. Missing References: Thank you for your advice. It is really close related to this work and will be included in revision.

---

### Official Review · Reviewer_nVw5 · 2023-08-11

**Soundness:** 4

**Excitement:**

4: Strong: This paper deepens the understanding of some phenomenon or lowers the barriers to an existing research direction.

**Paper Topic And Main Contributions:**

Previous contrastive learning methods suffers surface structure bias problem, often mistaking them for semantic similarity. As a result, paraphrased sentences with different surface structures might be deemed less semantically similar than sentences with merely a negative word insertion. This paper delves into this surface structure bias in unsupervised sentence embedding. By creating datasets that differentiate between surface and deep structure similarities, the authors present a two-fold solution: augmenting the data to counteract the bias with the max margin loss and utilizing recall loss to retain word semantics, thereby avoiding extreme forgetting.

**Reasons To Accept:**

- The paper presents an interesting unsupervised sentence embedding method based on contrastive learning. Notably, the authors first address the surface structure bias in unsupervised sentence embedding. They then employ a max margin loss to ensure proper ranking between negations and their paraphrased counterparts, and finally, they incorporate a recall loss to preserve knowledge from pretraining and minimize catastrophic forgetting.

- The empirical study is comprehensive, spanning various experiments from STS to SICK-R. The methodology's effectiveness has been tested on several architectures, including BERT base, BERT large, RoBERTa base, and RoBERTa large.

- The paper is well-written, with a clear delineation of its position relative to previous research. The emphasis on benchmarking against traditional baselines and surpassing them is commendable and will likely resonate with the broader community.


**Reasons To Reject:**

- Conceptually the proposed idea is similar to existing techniques and it could be viewed as incremental.
- The observed performance enhancements are somewhat modest, suggesting room for further refinement in the future.

**Reproducibility:**

3: Could reproduce the results with some difficulty. The settings of parameters are underspecified or subjectively determined; the training/evaluation data are not widely available.

**Reviewer Confidence:**

3: Pretty sure, but there's a chance I missed something. Although I have a good feel for this area in general, I did not carefully check the paper's details, e.g., the math, experimental design, or novelty.

---

> ### Author Rebuttal · Authors · 2023-08-29
>
> 1. About the contributions: Thank you for your comment. I agree with you that the study of a linguistic expression's surface structure and deep structure has long been proposed. However, since the unsupervised method based on contrastive learning has been proposed, this problem has not been revisited and verified whether it has been solved (as mentioned by the Reviewer w2Dv). This paper investigates the surface structure bias in contrastive learning for the unsupervised sentence representation and systematically evaluates the bias by constructing datasets following the two designated settings on surface structure and deep structure similarity, and then we propose a solution by leveraging augmented data with max margin loss and a catastrophic forgetting recall loss. Extensive experiments and analyses have shown that we significantly outperformed the baselines and demonstrated the effectiveness of our experiments.
>
> 2. About the improvement: Thank you for your comment. I assume our methods can continue to benefit from higher-quality synthetic examples thanks to the recent advancement of LLM models. We will leave it to future work.

---

### Meta-Review · Area_Chair_uJJ9 · 2023-09-18

**Recommendation:** 5

**Metareview:**

This paper addresses the surface structure bias issue in unsupervised sentence embedding models, particularly in contrastive learning methods. The authors propose a two-fold solution: first, they augment the data to counteract the bias, and second, they use a recall loss to preserve word semantics and avoid forgetting. This approach aims to improve the similarity assessment between sentences with different surface structures and sentences with semantic differences.
After author rebuttal, the reviewers updated their original views. In general, they agree that the paper is well-written, it's clearly motivated, and the empirical comparisons are comprehensive and sound. The reasons to reject are minors and the authors provided detailed comments that can added to the camera-ready version to enhance the paper.

---

### Decision · Program_Chairs · 2023-10-07

**Decision:**

Accept-Main

**Comment:**

This paper addresses the surface structure bias issue in unsupervised sentence embedding models, particularly in contrastive learning methods. The authors propose a two-fold solution: first, they augment the data to counteract the bias, and second, they use a recall loss to preserve word semantics and avoid forgetting. This approach aims to improve the similarity assessment between sentences with different surface structures and sentences with semantic differences.
After author rebuttal, the reviewers updated their original views. In general, they agree that the paper is well-written, it's clearly motivated, and the empirical comparisons are comprehensive and sound. The reasons to reject are minors and the authors provided detailed comments that can added to the camera-ready version to enhance the paper.